# Cancer Segmentation by Entropic Analysis of Ordered Gene Expression Profiles

**DOI:** 10.3390/e24121744

**Published:** 2022-11-29

**Authors:** Ania Mesa-Rodríguez, Augusto Gonzalez, Ernesto Estevez-Rams, Pedro A. Valdes-Sosa

**Affiliations:** 1The Clinical Hospital of Chengdu Brain Science Institute, University of Electronic Sciences and Technology of China, Chengdu 610054, China; 2Facultad de Matemática, Universidad de La Habana, San Lazaro y L, La Habana 10400, Cuba; 3Instituto de Cibernética, Matemática y Física, La Habana 10400, Cuba; 4Facultad de Física, Instituto de Ciencias y Tecnología de Materiales (IMRE), Universidad de La Habana, San Lazaro y L, La Habana 10400, Cuba; 5Centro de Neurociencias, BioCubaFarma, La Habana 10400, Cuba

**Keywords:** tumor discrimination, gene expression, Shannon entropy, information distance

## Abstract

The availability of massive gene expression data has been challenging in terms of how to cure, process, and extract useful information. Here, we describe the use of entropic measures as discriminating criteria in cancer using the whole data set of gene expression levels. These methods were applied in classifying samples between tumor and normal type for 13 types of tumors with a high success ratio. Using gene expression, ordered by pathways, results in complexity–entropy diagrams. The map allows the clustering of the tumor and normal types samples, with a high success rate for nine of the thirteen, studied cancer types. Further analysis using information distance also shows good discriminating behavior, but, more importantly, allows for discriminating between cancer types. Together, our results allow the classification of tissues without the need to identify relevant genes or impose a particular cancer model. The used procedure can be extended to classification problems beyond the reported results.

## 1. Introduction

Entropic magnitudes, as defined in information theory and related areas, have been used in a wide number of areas [1] beyond communication and physics, including literary analysis [2,3,4], painting [5], and music [6], among others. The advantage of entropic variables is the general framework on which they are founded, allowing them to be used in any system that conveys information in a broad sense. At the same time, it can be used to explore the emergence of patterns as opposed to noise as a random ordering of symbols without, in principle, resorting to a specific model of the data.

DNA is all about information, so there is no surprise in the use of information entropy to characterize the sequence of nucleotides [7,8,9,10] despite earlier works considering such analysis limited [11,12]. In this work, entropic magnitudes will be used as a tool for the classification of tissues between normal and tumor samples. Beyond its specific application in this contribution, the procedure presented shows that its generalization to other sequence analyses can be made straightforwardly.

More than two decades of experimental development have resulted in the availability of techniques to simultaneously probe thousands of gene expression levels from a single sample [13,14,15,16,17]. The access to such a huge data set has posed the challenge of meaningful harvesting of the information contained while discriminating, for a given purpose such as cancer diagnosis, the useful from the noise. Data analysis methods have been used over the years to feature-select and classify gene expression data [18,19,20,21]. Among others, this includes the use of statistical methods [22,23], neural networks and machine learning [24,25], and singular value decomposition-based methods [26,27]. In almost all cases, data analysis must effectively reduce the large feature set to identify a subset of distinctive genes that carry enough relevant information for the study at hand [28,29,30]. Without an underlying model, such reduction is still a challenge not robustly solved.

It is known that the identification of subtle patterns in gene expression can be used as the basis for classifying tissues, differentiating cancerous from non-cancerous samples [31,32,33,34,35,36]. This has been used not only for exploratory-diagnosis tasks but also to understand what transformation in cancerous cells is relevant and identify the genes responsible for it [15]. Recent studies have looked into the relationship between gene expressions and pathways in both coding and non-coding mutations of genes for a large number of cancer genomes [37]. The relevance of pathways can be essential to explore the cell circuitry and the cause of cancer. It can also prove helpful in designing new experiments with ad hoc perturbed cell conditions to test different hypotheses.

Entropic measures derive from several sources, starting with information theory [38] and including other fields such as Kolmogorov complexity [39], dynamical systems, and ergodic theory, among others. We applied entropic analysis to several samples from the Cancer Genome Atlas (TCGA) database for thirteen cancers. Two main results will be reported: our analysis, using complexity maps and information distance, shows strong discriminating behavior between tumor and normal tissue samples without the need for gene number reduction, avoiding sophisticated feature extraction procedures; the second result is clear evidence that ordering of the expression data along generic reaction pathways proves to be relevant to the cancer characterization in terms of organization and pattern production, a behavior that can not be explained on the sole basis of relative abundance of the different gene expressions.

## 2. Materials and Methods

### 2.1. Gene Expression Level and Gene Expression Coding

Gene expression data and their corresponding normal tissues are taken from the TCGA portal (https://portal.gdc.cancer.gov/) (accessed on 10 September 2020). RNA-Seq data in the FPKM format are used. The data contain expressions for 60,483 protein-coding, RNA genes, and pseudogenes. Gene expression levels are coded into three classes with a three-value alphabet χ∈{−1,0,1} as follows. The gene expression level is taken, and a value of 0.1 is added to each level allowing for geometric averaging, which is chosen due to the long tails in the gene expression distribution function. Geometric averaging is performed over the normal tissue set of samples, which allows the definition of a reference value for each gene expression, $e_{r}$. The differential expression is thus defined as $e/e_{r}$, and the base 2 logarithm is used to define a fold variation: $e_{f}=\log_{2}\left(e/e_{r}\right)$. We take $e_{f}<1$ as downregulated and assign a value of −1 to the gene expression coding; $e_{f}>1$ is taken as upregulated, and the value 1 is assigned to the gene coding; 0 is assigned otherwise.

### 2.2. Reactome Pathways and Ordered Sequences

For the coding of the biological pathways, the Reactome database was used. A list of pathways and the identified genes participating in them (https://reactome.org/download/current/Ensembl2Reactome.txt) (accessed on 5 February 2021) was ordered lexicographically. An ordered sequence of the gene codes was built by parsing the Reactome list and assigning to each entry the corresponding code (See Figure A1 in the Section A.1).

### 2.3. Entropic Measures

Consider a bi-infinite sequence *s* of values $s_{i}$ at position *i* (−∞<i<∞), taken from a discrete alphabet χ. The (Shannon) block entropy of length *L* of the sequence is defined as $H\left(L\right)=\sum_{s\left(L\right)\in \chi^{L}}p\left[s\left(L\right)\right]\log p\left[s\left(L\right)\right]$, where p[s(L)] is the probability of finding a finite subsequence s(L) of length *L* in the bi-infinite string *s*. There are ${\left|\chi \right|}^{L}$ possible finite sequences of length *L*. If in the definition of block entropy, the logarithm is taken in base 2, the units are bits. The entropy density is then $h=\lim_{L\to \infty }H\left(L\right)/L$; it measures the entropy per symbol when an infinite number of symbols have been observed, and therefore the irreducible randomness per symbol of the sequence. The entropy density is zero for a constant, a periodic, and a quasi-periodic string, while it attains its maximum value log|χ| for a completely random sequence.

Consider h(L)=H(L)−H(L−1), the effective measure complexity (*E*) [40], also known as excess entropy [41], can be defined as $E=\sum_{L=1}^{\infty }\left[h(L)-h\right]$. *E* is the mutual information between the two halves of the bi-infinite string [1,42]. It measures the memory or amount of information of one half of the sequence carried to the other half. We will be using plots of *E* vs. *h*, which is an example of what is called a complexity–entropy map. Complexity–entropy maps show the trade-off between randomness or pattern production as measured by *h* and pattern recurrence or memory as measured by *E*; it has been used (under different names) in studies of an extensive set of systems [1,4,43].

Finally, information distance between two sequences comes from Kolmogorov complexity theory [44], and the Kolmogorov complexity K(s) of a string *s* is the length of the shortest program capable of reproducing the string when run in a Universal Turing Machine [39]. A pattern recurrence-driven sequence will have a Kolmogorov randomness that grows slowly (e.g., as log(L) ) with the sequence length, such that its normalized value K(L)/L→0 as L→∞. The same ratio goes to 1 for a random sequence. Information distance between two sequences *s* and *q* is defined as $d\left(s,p\right)=\left[K(sq)-min\{K(s),K(q)\}\right]/max\left\{K\left(q\right),K\left(s\right)\right\}$, where sp represents the concatenation of both strings. Information distance measures how innovative a string is with respect to the other from an algorithmic perspective: it measures the shortest program length that can transform one string into the other. Information distance has been used to study the relationship between instances of a system in a wide set of areas, for example, to extract phylogenetic evolutionary trees from the genes of mammals [39].

All entropic measures are estimated using the Lempel–Ziv approximation.

### 2.4. Lempel–Ziv Estimates

When finite data are considered, the entropy rate has to be estimated [42,45] as the asymptotic and limiting values are unreachable. The Lempel–Ziv factorization [46] procedure has been used for the estimation of the entropy density.

Consider the following factorization of a sequence $s=s_{1}s_{2}\ldots s_{N}$:$$F\left(s\right)=s\left(1,p_{1}\right)s\left(p_{1}+1,p_{2}\right)\ldots s\left(p_{j-1}+1,p_{j}\right)\ldots s\left(p_{m-1}+1,N\right),$$
where $p_{k}=1,2,\ldots$ is a natural number, and s(p,q) is the substring $s_{p}s_{p+1}\ldots s_{q}$. Each symbol $s_{p}$ is drawn from a finite alphabet χ of cardinality |χ| (e.g., $\chi =\{\bar{1},0,1\}$, |χ|=3).

F(s) is called an exhaustive history of the sequence *s*, if any factor $s(p_{j-1}+1,p_{j})$ is not a substring of the string $s(1,p_{j}-1)$, while $s(p_{j-1}+1,p_{j}-1)$ is, except perhaps for the last factor.

For example, the sequence $01\bar{1}01\bar{1}100\bar{1}\bar{1}01010011111\bar{1}\bar{1}\bar{1}\bar{1}00\bar{1}10101111$, taken from an alphabet $\chi =\{\bar{1},0,1\}$, has the following factorization, where a dot separates each factor:$$0\cdot 1\cdot \bar{1}\cdot 01\bar{1}1\cdot 00\cdot \bar{1}\bar{1}\cdot 010\cdot 1001\cdot 111\cdot 1\bar{1}\bar{1}\cdot \bar{1}\bar{1}00\cdot \bar{1}1\cdot 01011\cdot 11$$

The Lempel–Ziv complexity C(s) is then the cardinality (number of factors) of the exhaustive history F(s) (In the above example, C(s)=14).

In general, C(s) for a length *N* string is bounded by [46]
(1)$$C\left(s\right)<\frac{N}{\left(1-\varepsilon_{N}\right)\log_{\left|\chi \right|}N},$$
where
(2)$$\varepsilon_{N}=2\frac{1+\log\log(|\chi |N)}{\log N}.$$

We used $\log x\equiv \log_{\left|\chi \right|}x$ to simplify the notation. $\varepsilon_{N}$ is a slowly decaying function of *N*, leading to an asymptotic value
(3)$$C\left(s\right)<\frac{N}{\log N},$$
for large enough *N*.

Ziv [47] proved that, if *s* is the infinite length output from an ergodic source with entropy rate *h*, then
(4)$$\underset{N\to \infty }{\mathrm{lim sup}}\frac{C[s(1,N)]}{N/\log N}=h.$$
almost surely. The use of the Lempel–Ziv factorization for estimating the entropy density for finite-size sequences has proved robust even for short-length strings [45]. For a $10^{4}$ length sequence, which will be used in this study, the order of magnitude for the error bound is around $10^{-2}$ [48]. The Lempel–Ziv factorization procedure was implemented in an in-house software (written in C++ and with run time below one minute for each data set) and has been used in previous studies [4,49,50].

The effective complexity measure *E* is estimated using a random shuffle procedure given by [51]
(5)$$E\left(s\right)=\sum_{M=1}^{M_{max}}\left[h\left(s_{\left(M\right)}\right)-h\left(s\right)\right].$$
$s_{\left(M\right)}$ is a surrogate string obtained by partitioning the string *s* in non overlapping blocks of length *M* and performing a random shuffling of the blocks. The shuffling for a given block length *M* destroys all correlations between symbols for lengths larger than *M* while keeping the same symbol frequency. $M_{max}$ is chosen appropriately given the sequence length to avoid fluctuations. In spite of the fact that Equation (Equation 5) is not strictly equivalent to the *E*, it is expected to behave in a similar manner [51].

As already explained, information distance d(s,p) comes from the use of algorithmic randomness [52] $K\left(s\right)=|s^{*}|$ of a string *s*, or the length of the shortest algorithm $s^{*}$ capable of producing the string *s* when run in a Universal Turing Machine.

It is known [39] that the entropy density *h* is also given by
(6)$$h\left(s\right)=\lim_{|s|\to \infty }\frac{K(s)}{\left|s\right|}.$$

From this result, it follows that [49]:(7)$$d\left(s,p\right)=\frac{h(sp)-min\{h(s),h(p)\}}{max\{h(s),h(p)\}}.$$

Again, we estimate the entropy density via Lempel–Ziv factorization and, from there, d(s,p).

## 3. Results

Gene expression data for 13 tumors were studied (a table with the TCGA nomenclature for the cancers studied is published as Appendix A
Table A1). The data contain expressions for 60,483 protein-coding, RNA genes, and pseudogenes. Gene expression levels are coded into three classes with a three-value alphabet $\chi \in \{\bar{1},0,1\}$ corresponding to under-, normal-, and over-expressed, respectively. The list of gene expression classes is ordered using a list of pathways and the identified genes participating in them. Such a list can be considered a data sequence.

### 3.1. Complexity–Entropy Maps

For every cancer type, the entropy and *E* were estimated from a randomly ordered sequence resulting from the unordered nature of the gene expression data. The corresponding complexity–entropy map was plotted, as shown for the COAD data in Figure 1a. The cancer samples tend to have higher values of entropy density. While segmentation is possible using the entropy density value, no relation was found with the *E* values. Upon sorting the gene expression classes with the pathway list, a different picture emerges as *E* strongly correlates with the *h* value. Figure 1b shows that segmentation is now possible, considering both the *h* and the *E* value. For the samples of cancer tissue, there is an order of magnitude increase of *E* compared to the unsorted strings, which points to the increasing appearance of structuring in the sorted data set.

From now on, all results are referred to the sorted data.

Normal tissue exhibits a mean (median) *h* value of 0.304 (0.251) (see Table 1) while, for the tumor sample, the mean (median) value is 1.012 (1.013), a 3.32 (4.04) fold increase. For *E*, the increase for the mean (median) is 3.85 (4.43) times from 0.537 (0.460) to 2.067 (2.037). The Euclidean distance in the complexity–entropy map between the mean(median) value of the normal sample and the tumor sample was calculated to be 1.686(1.751). Similar results were found in other cancer types.

Structuring may come from two processes: change in the symbols fractions, in this case, the symbols represent one of three classes for up- and down- and normal-regulated genes, so this process refers to the change in the symbol fraction in each class; on the other hand, structuring can be the results of the ordering of the symbols. Structuring by symbol fraction change usually dominates when one of the symbols becomes dominant in the data.

Tumor samples show a larger number of up- and downregulated genes at the other genes’ expense than the normal samples. Figure 1c shows a change in symbols fraction when the normal samples with the tumor samples are compared. However, when we plot the complexity–entropy map of the difference between the unsorted set values and the sorted set values (ΔE vs. Δh, where $\Delta E=E_{sorted}-E_{unsorted}$ and similar for Δh), as shown in Figure 1d, it becomes evident that the increase in *E* is mainly the result of symbol rearrangement, as any change coming from symbol fraction changes is canceled. The same can be said for the increase of *h* values.

A straightforward discrimination procedure was used based on the complexity–entropy map. The Euclidean distance to the mean (median) values of the normal and tumor points was calculated for every sample. The class to which a sample belongs was decided by taking the one with the shortest Euclidean distance from its mean (median) value to the sample point. With such criteria, the success rate of detecting normal tissue was 0.976 (0.976) and of detecting tumor tissue was 0.979 (0.983).

Similar analyses were performed for the rest of the cancer types, and the complexity–entropy map for each is published as Appendix A. Table 2 summarizes the results. While good discriminating results are robust, above 0.9 for most cancer types, the present procedure had a lower performance for the PRAD, STAD, and THCA cancer types. The worst discriminating fraction was that of the PRAD samples with values around 0.7.

If the complexity–entropy map for the mean value of all cancer types is plotted (Figure 2), two groups can be identified. One on the upper right of the plot corresponds to the tumor tissue data, exhibiting larger *h* and *E* values. The second group corresponds to the normal tissues at the lower-left corner with smaller *h* and *E* values. The poor discriminating performance of the PRAD, STAD, and THCA samples comes from the fact that their mean values for cancer (PRAD, THCA) and normal (STAD) tissue data can not be included in their corresponding groups.

### 3.2. Discrimination by Information Distance Measure

The information distance matrix for all samples within a cancer type was computed; we show in Figure 3 the dendrogram corresponding to the COAD tumor type (the dendrogram for all cancer types is published as Appendix A).

From the dendrogram of Figure 3, it is clear that the normal and tumor tissue are clustered in the plot hierarchy. Interestingly enough, all the branches of the cancer tissue show deeper levels of derivation from the root. The result could point to further exploring the structure of the dendrogram in terms of the cancer gene expression in the individual tissues. Discrimination of normal and tumor tissues can be made for the dendrogram alone, yet we used a different approach.

For the distance matrix, a discriminating procedure was designed based on a majority rule. The information distance to the other tissue samples was considered for each sample, and the nearest neighbor type was used as a majority discriminating criterion. The sample was assigned to the type, tumor, or normal, based on the majority of neighbor types (Figure 4). For each cancer type, the number of neighbors to consider was taken between 3 and 8 and optimized for performance.

Consider a neighborhood formed by the eight nearest samples. For the COAD type cancer, if we assigned a value of −1 for each cancer neighbor and a value of 1 for each normal neighbor, the average number of neighbors for a random sample is −6.94, showing that there are many more cancer samples than normal samples. However, the average neighbor number for the normal sample is 5.64, pointing to the fact that normal samples surround normal samples; and the average number for the cancer sample is −7.98, showing that cancer samples surround cancer samples.

A second discriminating procedure was also used, where the distance weights each neighbor, and no significant difference was found.

The success fraction for all cancer types is reported in Table 3. While the success rate for the normal tissue is lower in the PRAD case, compared to the one using Euclidean distance in the complexity–entropy map, the success rate for the cancer tissue is much higher, up to 0.994, similar values to those obtained for the higher ranking success rates in Table 2. The smaller success fraction for the cancer tissue was as high as 0.973 for the KIRP sample. Care must be taken, though, with such a high success fraction, as the number of the normal tissue samples and tumor samples is biased towards the tumor sample, which results, statistically, in a larger number of tumor neighbors for the random case. The same comment can be made on the low success rate for the normal tissue, as, statistically, there are fewer normal neighbors than tumor type. Improvement should be expected when more normal tissues are included in the sample set.

Finally, it was studied whether using the distance matrix allows for discriminating between types of cancer. The KIRC and KIRP kidney cancer were chosen for two reasons. On the one hand, they are two types of tumors from the same organ; therefore, discrimination from the genetic data extracted from the sample tissue could make clinical sense. On the other hand, both mean value points are too close to consider discrimination from such analysis in the complexity–entropy map. Expression reference values were calculated using the set of normal tissue values for both the KIRC and KIRP datasets. The expression classes were determined as already explained using these common reference values. The distance matrix was calculated between all samples from both tumors, KIRP and KIRC, and the normal tissues. The distance matrix and dendrogram are shown in Figure 5, where each cancer type and the normal sample are grouped in mostly separate clusters. Using the discrimination procedure described above, 0.998 of cancer tissues were identified as a tumor, and from there, 0.939 were correctly identified for KIRC type tumors and 0.913 as KIRP type cancer. In addition, 0.981 of normal tissues were correctly identified as such. The same analysis was carried out for the LUSC and LUAD cancer types. The success fraction for cancer tissue was 0.994. The 0.839 and 0.984 fractions of LUAD and LUSC cancer were correctly discriminated, respectively. The dendrogram is shown in Section A.5.

## 4. Discussion

This work reports the application of entropic measures that proved useful in analyzing an extensive gene expression data set. The measures seem to allow, at least for the scope of this study, the segmentation of tumor tissue samples from normal tissue samples for various cancer types. The sorting of the data along pathways enhanced the segmentation abilities of the applied techniques. One may ask if this points to features beyond cancer discrimination. Gene expression data studies from pathway analysis have proven that, even in the case where no significant variation of a single gene expression level is involved, studying groups of genes by their functional role identifies the difference between samples of healthy subjects and subjects with some types of diseases or abnormal condition, such as diabetes and smoking epithelial tissues [27,54]. Such studies emphasize that the analysis of gene expression levels alone may be insufficient to recognize or characterize a disease condition. In the hallmark paradigm of cancer, [55,56], changes in gene expression in tumor cells results in the reprogramming of the cell circuitry to sustain the so-called hallmarks of cancer. The intracellular integrated circuitry is divided into distinct subcircuitry, each of which performs some specialized function in normal cells, which is modified when changed to cancer cells. In this picture, the effect of varying gene expression levels is relevant as it modifies specific pathways supported by the cell circuitry. Our studies emphasize such a picture, as the pathway analysis of gene expression profiles, via entropic measures can be interpreted as grasping some of the relevant features of this circuitry modification.

Entropic measures have been used to study the pattern gain at different levels of grammar hierarchy in written language [4]. In our case, the ordered sequence has two possible organization levels, one within the pathways and the other from the pathways position in the list. In both levels, there is not a priori “natural” ordering, and lexicographic order, as the one used, is, in principle, as good as any other ordering scheme. However, the ordered data immediately show the emergence of a clear trend between entropy density and *E*. The simultaneous increase of entropy density and *E* is a fingerprint of increased complexity. As measured by *h*, the given disorder can still accommodate enough pattern recurrence measured by *E*. In this sense, the analysis of the tumor samples shows a more complex pathway-ordered gene expression profile. This complexity does not arise mainly by changes in the fraction of samples in each symbol class, although it is clear that up- and downregulated genes increase at the expense of the non-regulated ones. The role of gene expression levels in connection with cancer’s appearance is usually discussed, but what is pointed out here is that this does not describe the whole story. There is no real physically meaningful order for the pathways, so randomization breaks any pattern at the intra-pathway ordering level, leading immediately to the disappearance of any relation between pattern production and pattern recurrence as *E* drops an order of magnitude.

## 5. Conclusions

We have proven that, in several cases, and within the scope of the present study, entropy measures allow the discrimination of cancer samples from normal samples by a class analysis of gene expression levels. In both complexity–entropy maps and information distance, classification was possible with a high success rate within a cancer type, above 98% for several cancer types. Furthermore, information distance allows for better discrimination when more than one cancer type is involved. The studies in two types of kidney tumors allowed the correct identification of the tumor type with a very high success rate. The study also suggests that increasing the number of samples, as seen in the joint analysis of KIRC and KIRP data, enhances the classification procedure’s robustness. The emergence of a clear trend in the complexity–entropy map upon sorting the gene expression level through an ordered pathways list allowed us to identify that, as crucial as upregulated and downregulated genes are, the whole context needs to look into the pathways that are activated or turned off in cancer samples.

Further experimental work is needed to evaluate if our findings are relevant in practical tumor discrimination, and current work is planned along this line. In addition, the reported results point to expanding the method to explore if it can be used to study different stages in cancer development and the occurring changes along its evolution.

The developed procedure can be extended to other data types as its generalization is straightforward.

## Figures and Tables

**Figure 1 entropy-24-01744-f001:**
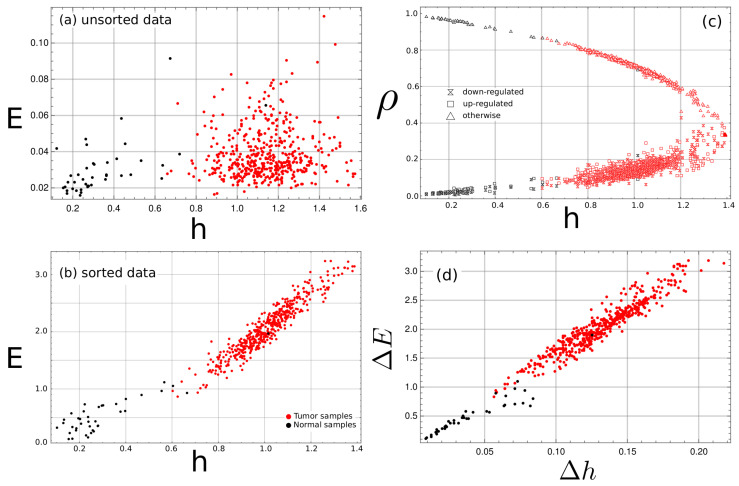
**Complexity–entropy map for the COAD tumor type tissue samples.** Each point corresponds to a tissue sample. Red points are tumor samples, while black points correspond to normal samples. In (**a**,**b**,**d**), the vertical axis represents the *E* values, while the horizontal axis represents the entropy density. Three classes are considered up-, down- and normally expressed genes. In (**c**), the vertical axis represents the fraction of samples in each class. The entropy map *E* vs. *h* of the unsorted data (**a**) fails to show any correlation between the entropy rate *h* and the *E* values; upon sorting by pathways (**b**), a clear trend emerges in the map where larger values of *h* imply larger values of *E*. (**c**) The fraction ρ of the number of tissues in a given class, with respect to the total number of samples, shows that tumor samples have a larger number of down- and up-expressed genes compared to the normal samples, but the difference between the unsorted and sorted complexity–entropy map $\Delta E(=E_{sorted}-E_{unsorted})$ vs. $\Delta h(=h_{sorted}-h_{unsorted})$ (**d**) demonstrate that the increase of *E* values of the tumor tissue can not be explained by a change in symbol fraction only, but also by pattern formation and ordering.

**Figure 2 entropy-24-01744-f002:**
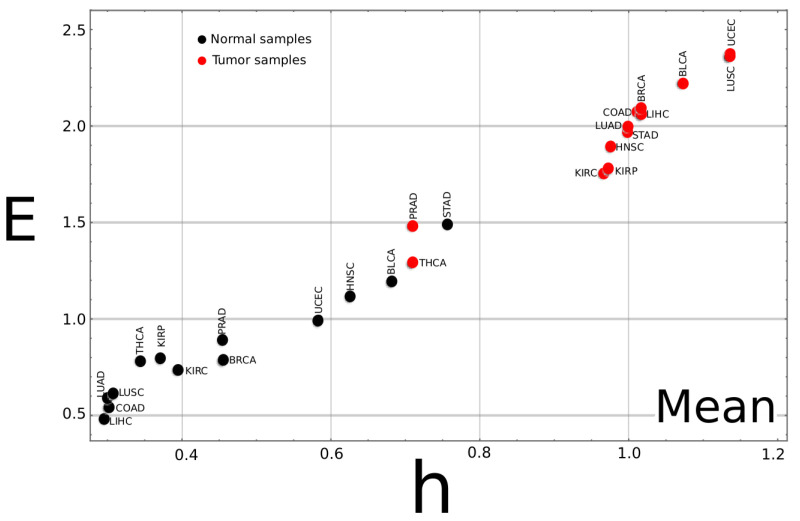
**Complexity–entropy map for the mean values of all cancer types.** The mean value for the two classes, tumor and normal, was calculated for each of the 13 tumor types. As a general trend, the plot shows that tumor samples exhibit larger *h* and *E* values than normal tissues. It can be noticed that, for the STAD type cancer, the (h,E) point for the normal sample is unusually high compared to the other samples type, while the points for tumor samples of the PRAD and THCA samples are much lower than the other cancer types. These three cancer types exhibit the worse success ratios (a similar plot for the median value can be found as Appendix A).

**Figure 3 entropy-24-01744-f003:**
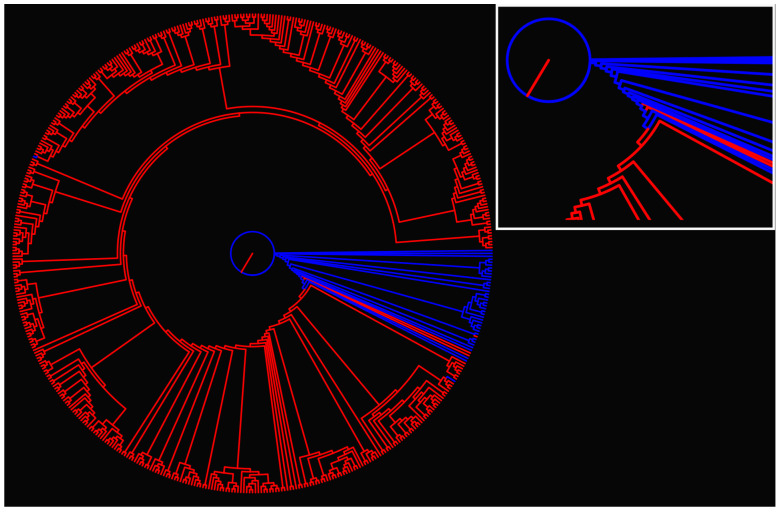
**Dendrogram for COAD type cancer.** Information distance d(s,p) between all samples for a given cancer type was calculated, resulting in a distance matrix. The dendrogram is built from the distance matrix using the Phylip suit of programs [53]. Blue color corresponds to normal, and red to tumor tissue data. The inset shows the details at the root of the dendrogram.

**Figure 4 entropy-24-01744-f004:**
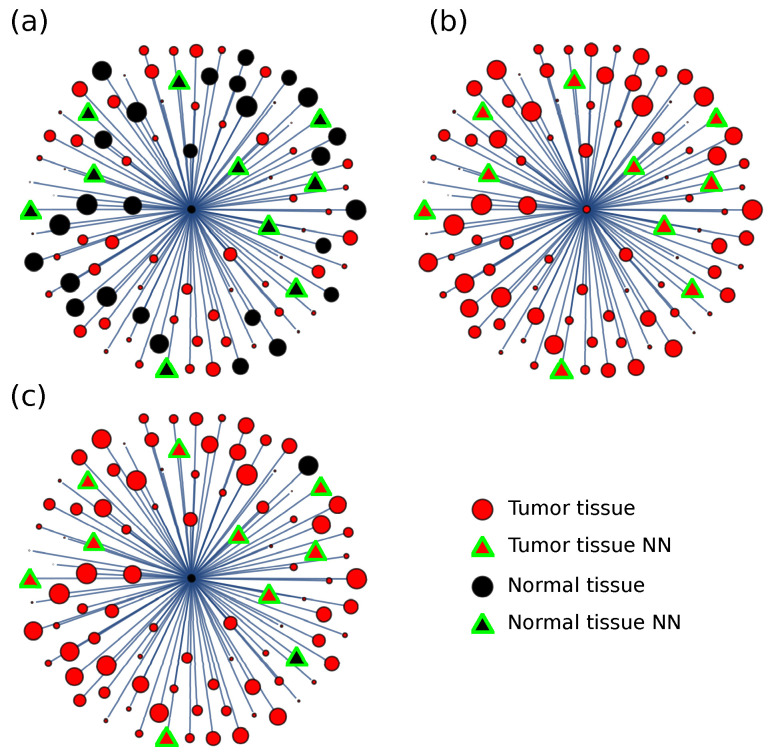
**Information distance discriminating criteria.** Given a sample (at the center), the distance d(s,p) to any other sample is calculated, and the nearest neighbors (NN) are considered for discriminating purposes. The sample is classified in the (**a**) normal or (**b**) tumor class, depending on the class with majority NN (distances in figure not at scale). For the COAD type cancer, in seldom cases, the used criteria (**c**) failed to ascribe the correct class to the sample.

**Figure 5 entropy-24-01744-f005:**
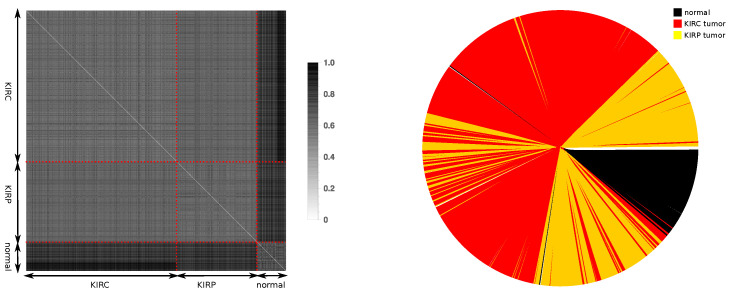
(**left**) Distance matrix and (**right**) information distance dendrogram for the joint KIRC and KIRP tumor types together with the normal tissues. For the fold variation of each gene, the $e_{ref}$ values were calculated from the joint set of normal tissue samples from both the KIRC and KIRP datasets.

**Table 1 entropy-24-01744-t001:** **Mean and median values of the entropic measures for tumor and cancer tissue samples.***h* stands for the entropy density, *E* for the effective measure complexity, and *d* is the Euclidean distance between the normal and tumor sample’s mean (median) values. The table includes the values for all cancer types studied.

	Mean					Median				
	Tumor		Normal			Tumor		Normal		
	*h*	*E*	*h*	*E*	*d*	*h*	*E*	*h*	*E*	*d*
**BLCA**	1.073	2.222	0.683	1.192	1.101	1.098	2.260	0.730	1.178	1.143
**BRCA**	1.017	2.097	0.456	0.785	1.426	1.045	2.169	0.408	0.722	1.581
**COAD**	1.012	2.067	0.304	0.537	1.686	1.013	2.037	0.251	0.460	1.751
**HNSC**	0.975	1.891	0.626	1.119	0.847	0.985	1.864	0.583	1.016	0.939
**KIRC**	0.967	1.754	0.395	0.733	1.170	0.941	1.706	0.371	0.705	1.152
**KIRP**	0.973	1.772	0.372	0.797	1.145	0.950	1.682	0.384	0.790	1.057
**LIHC**	1.018	2.060	0.295	0.480	1.738	1.057	2.138	0.249	0.469	1.854
**LUAD**	0.999	1.964	0.301	0.581	1.549	1.017	1.946	0.259	0.584	1.559
**LUSC**	1.137	2.382	0.308	0.611	1.955	1.172	2.439	0.291	0.620	2.021
**PRAD**	0.712	1.481	0.454	0.896	0.639	0.700	1.435	0.394	0.796	0.708
**STAD**	1.000	1.989	0.757	1.486	0.558	1.031	2.010	0.730	1.449	0.636
**THCA**	0.710	1.296	0.346	0.780	0.632	0.691	1.246	0.301	0.678	0.689
**UCEC**	1.138	2.382	0.585	1.002	1.486	1.144	2.378	0.589	1.012	1.474

**Table 2 entropy-24-01744-t002:** **Success fraction using Euclidean distance as discriminating criteria in the complexity–entropy map.** The mean and median values for the normal and tumor points were calculated. A given sample is ascribed to the class whose (h,E) point has the smaller Euclidean distance to its mean (median) value. Each sample from a given tumor type was used as a test sample to compute the success ratio. The fraction corresponds to the ratio between the number of correctly classified samples and the total number of samples in a given class.

	Mean		Median	
	Cancer	Normal	Cancer	Normal
**BLCA**	0.917	0.947	0.903	0.947
**BRCA**	0.912	0.937	0.912	0.937
**COAD**	0.979	0.976	0.983	0.976
**HNSC**	0.880	0.750	0.934	0.750
**KIRC**	0.983	0.958	0.991	0.958
**KIRP**	0.979	1.000	0.990	1.000
**LIHC**	0.933	1.000	0.930	1.000
**LUAD**	0.933	0.983	0.938	0.983
**LUSC**	0.972	1.000	0.966	1.000
**PRAD**	0.681	0.731	0.735	0.731
**STAD**	0.723	0.656	0.723	0.656
**THCA**	0.872	0.914	0.916	0.879
**UCEC**	0.987	1.000	0.985	1.000

**Table 3 entropy-24-01744-t003:** **Success fraction using the information distance as discriminating criteria.** From the distance matrix, the nearest neighbors for a given sample are taken, and a majority rule is used to classify the sample as a tumor or normal type. The fraction corresponds to the ratio between the number of correctly classified samples and the total number of samples in a given class.

	Cancer	Normal
**BLCA**	0.997	0.210
**BRCA**	0.995	0.821
**COAD**	1.000	0.872
**HNSC**	0.996	0.454
**KIRC**	0.998	0.972
**KIRP**	1.000	0.969
**LIHC**	0.973	0.760
**LUAD**	0.994	0.847
**LUSC**	0.998	0.959
**PRAD**	0.994	0.404
**STAD**	0.987	0.719
**THCA**	0.986	0.845
**UCEC**	0.996	0.609

## Data Availability

Gene expression data were extracted from the TCGA portal 2020[2020-09-10](https://portal.gdc.cancer.gov/). Processed data can be downloaded at Science Data Bank, 2022[2022-11-18]. https://cstr.cn/31253.11.sciencedb.06470.CSTR:31253.11.sciencedb.06470. An in-house program was used for Lempel–Ziv estimation. E.E-R. can be contacted for availability. Plots were made using Wolfram Mathematica (R) framework.

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
