# Peer review of "Cancer Segmentation by Entropic Analysis of Ordered Gene Expression Profiles"

_entropy, 2022, doi:10.3390/e24121744_

Round 1
Reviewer 1 Report
I like the manuscript and the idea of the work. The results are nice. The results part is, however, hard to understand. The figures and results are not well explained. I don’t see the reason to present phylogenic trees (dendrogram, Figures 3, 5b, A4, A5) if you don’t make any use of them. In some cases, the authors designed their own methods. Their design of methods lacks a consequent strategy and interpretation of the outcomes is not always obvious.
There exists a bunch of famous standard methods of bisection, clustering, and modularization of networks. There exits famous standard methods of binary classification, e.g., in the field of machine learning. The authors either ignore the standard methods or are not aware of them. The presented work may profit from a co-operation with a computer scientist or bio-informatician to improve the manuscript from the viewpoint of methods. Note that, standard methods are implemented in a number of available packages, as, e.g, scikit-learn, radatools and many more. The excellent review of Fortunato, Santo. "Community detection in graphs." Physics reports 486.3-5 (2010): 75-174 may give some orientation.
I would like to encourage the authors to revise the results part. Please, lead the reader through your tables and figures. Explain what the figures show. Note that, the reader see the figures the first time and is not familiar what red/black scatter points represent. During reading the manuscript, I became more and more frustrated. My remarks below may help to improve the readability of the manuscript.
The conclusion of the authors that, sorting of the data along pathways enhanced the segmentation abilities is based only on visual inspection of figures. It may be worthwhile to prove the statement by statistics and numerical tests. A simple statistical Mann-Whitney U test may be applied to compute the AUC and significance of entropy rate h and excess entropy E. A comparison of the AUC values with and without pathway ordering would be conclusive. You may apply a simple Random Forrest to show that, E and h gain predictive power if the sequence is ordered according to pathways.
The manuscript refers to supplementary material with numbers as, e.g., Figure S1. You have provided no supplementary material as Figure S1. Please, synchronize the references to appendix/supplement.
Software: Please clarify. How you have produced the results and figures? What software have you applied? Have you implemented the calculations? What language? Any use of packages? Add a comment on the run time.
Reference values were computed by geometric averaging after adding a value of 0.1.
The choice of the addition of a value 0.1 is confusing.
How have you chosen the value 0.1?
In case of only zero gene expression in normal tissue, you get a reference value of 0.1. In cancer tissue, gene expression below the arbitrary value 0.1 is classified as "down regulated" (value -1), gene expression above 0.1 is classified as upregulated" (value +1). An arbitrary regulation constant (0.1) defines were you set the threshold to distinguish between up and down regulated proteins.
How would the results change if you apply an addition by a larger value, as, e.g., 100?
You should justify the choice of the value 0.1 (e.g., in comparison to mean/median gene expression value).
For long tail distributions, the median value may be a more conventional choice to compute a stable reference values.
Please provide some statistical characterization of the data, e.g.:
How many genes do you have in the classes’ χ ∈ {−1, 0, 1}?
How many proteins are repeated n=0, 1, 2, 3 ... times in the pathway list?
Please, check equations in text from line 99 to line 107. There are some typos. Introduce unique terms and symbols for the relevant concepts. Use introduced terms and symbols through the entire manuscript, e.g., entropy rate and entropy density should not synonymously be used.
Page 5, sentence "A table with the TCGA nomenclature for the cancers studied is published as supplementary material". Please specify the Table number.
Figure 1. Please describe the figure in the caption. How many points have the scatter plot? What data set is used? What does a point represent? What are the meanings of blue lines? The figures should become comprehensive by reading the caption.
What is shown in Figure 1c? How do you compare the normal samples with the tumor samples? What represents a point? What is the fraction of the number of tissues in a given class? What classes do you have? The reader can only guess what you have plotted in Figure 1c.
I don't get the reason for the discussion of the effect of "production of new symbols". You observe an increase in EMC value by an order of magnitude if you reorder the sequence. The source of the difference is obviously the order and not the production of new symbols. The reordering have produced redundancy (compare Figure 1a and 1b), no new symbols have been produced to compute Figure 1b. Please, clarify what effect concerns the discussion of the production of new symbols?
Table 3. What algorithm was applied to bisect the tissue samples? "The class to which a sample belongs was decided by taking that of the shortest distance." Shortest distance to what values? Mean, median or something other?
Section: Discrimination by information distance measure. Figure 3. What cluster method was applied? Since you are interested only in bisection, the representation as phylogenetic tree is not very instructive. You may want to delete figures as, e.g., Figures 3, A4, and A5.
The idea of the discriminating procedure that was designed by the authors is confusing. If we know the class of each of its neighbors, we can classify the tissue. The number of neighbors is optimized to get the best result. In don't see how such a procedure can help to classify. The authors ignore the broad variety of available methods for bisection, as, e.g., k-means, and Kernighan-Lin, we would refer to the excellent review of Fortunato, Santo. "Community detection in graphs." Physics reports 486.3-5 (2010): 75-174.
Reviewer 2 Report
See attached file.

Round 2
Reviewer 1 Report
Tenses may still be a small issue. Standard usage of tenses would help the reader to distinguish between results of the authors (past tense)
and previous studies (present perfect), e.g., page 2, the sentence
"We have applied entropic analysis to several samples from The Cancer Genome Atlas (TCGA) database for thirteen different cancers."
may be easier to read in past tense:
"We applied entropic analysis to several samples from The Cancer Genome Atlas (TCGA) database for thirteen different cancers."
Reviewer 2 Report
Thanks, for the revised version of the manuscript. The manuscript has been improved, but there are still points:
- there are multiple abbreviations for Effective Measure Complexity: E and EMC?
- there are still typos e. g. Line 220 “a 1” ??? and English can still be improved. I recommend to have the manuscript checked by a native speaker.
